# Joint Effect of Diabetes Mellitus and Hypertension on COVID-19 in-Hospital Mortality Stratified by Age Group and Other Comorbidities: A Cohort Retrospective Study Using Hospital-Based Data in Sleman, Yogyakarta

**DOI:** 10.3390/healthcare10102103

**Published:** 2022-10-20

**Authors:** Yampa Eksa Daidella Ghilari, Arik Iskandar, Bayu Satria Wiratama, Anggoro Budi Hartopo

**Affiliations:** 1Master of Public Health Program, Faculty of Medicine, Public Health, and Nursing, Universitas Gadjah Mada, Yogyakarta 55281, Indonesia; 2Department of Biostatistics, Epidemiology, and Population Health, Faculty of Medicine, Public Health, and Nursing, Universitas Gadjah Mada, Yogyakarta 55281, Indonesia; 3Graduate Institute of Injury Prevention and Control, College of Public Health, Taipei Medical University, Taipei 110, Taiwan; 4Department of Cardiology and Vascular Medicine, Faculty of medicine, Public Health, and Nursing, Universitas Gadjah Mada, Yogyakarta 55281, Indonesia

**Keywords:** diabetes mellitus, hypertension, cohort retrospective, logistic regression, joint effect, COVID-19, mortality

## Abstract

The joint effect of diabetes mellitus and hypertension on COVID-19 has rarely been evaluated but had potential as a major risk factor. This study aims to investigate the joint effect between diabetes mellitus and hypertension on in-hospital mortality among COVID-19 patients in Yogyakarta stratified by age groups and other comorbidities status. **Methods:** This cohort retrospective study collected data from two major hospitals in the Sleman district and a total of 2779 hospitalized COVID-19 patients were included in this study. The study outcome was COVID-19 in-hospital mortality (deceased or discharged alive) and the main risk factors were diabetes mellitus (DM) and hypertension (HT). The multiple logistic regression model was utilized to estimate adjusted odds ratio (AOR) and calculate the joint effect. **Results:** COVID-19 patients who have both DM and hypertension were three times (AOR: 3.21; 95% CI: 2.45–4.19) more likely to have in-hospital mortality than those without both comorbidities. The highest risk of in-hospital mortality was found in COVID-19 patients without other comorbidities (other than DM and HT) and younger age (age 0–40 years), with AOR equal to 22.40 (95% CI: 6.61–75.99). **Conclusions:** This study identified a joint effect between diabetes mellitus and hypertension which increases the risk of in-hospital mortality among COVID-19 patients. Targeted public health, clinical, and health education intervention should be carried out on individuals with diabetes mellitus and/or hypertension.

## 1. Introduction

SARS-CoV-2 is a virus that causes infections in the respiratory tract, which can also cause severe lung infections and even death [1,2]. The prevalence of COVID-19 cases in Indonesia ranks 14th worldwide as of 29 December 2021, with 4,262,351 positive cases and 144,081 deaths [3]. The Special Region of Yogyakarta itself, as of 30 December 2021, ranks the sixth most cases in Indonesia, with a cumulative number of 156,991 positive cases and 5269 deaths. The prevalence of hypertension (HT) in DIY, according to Riskesdas (2018), is 11.01% or higher compared to the national figure (8.4%), making DIY rank fourth as a province with high hypertension cases [4]. Meanwhile, diabetes mellitus (DM) is around 4.8%, higher than the national figure of 1.8, meaning Yogyakarta has the third-highest number of diabetes cases [4].

People with confirmed NCDs and COVID-19 have a high risk of clinical deterioration and early mortality [2,5,6,7], especially those with confirmed COVID-19. Those with diabetes mellitus in COVID-19 have a mortality rate that is 12.23 times higher than those without diabetes mellitus [5,6,7,8]. COVID-19 patients with hypertension were more likely to have in-hospital mortality than those without hypertension [9]. A meta-analysis conducted by Raymond et al. [10] showed that hypertension is a risk factor of COVID-19 mortality and severe COVID-19. The mortality rate for COVID-19 individuals with DM was found to be 1.4% higher in an American study [11].

The joint effect of diabetes mellitus and hypertension on COVID-19 has rarely been evaluated but had potential as a major risk factor. Research conducted in Brazil showed that the joint effects between age >65 with diabetes mellitus and hypertension were significant risk factors on COVID-19 mortality [2]. Another study in China found that there is a joint effect between diabetes mellitus and hypertension, even though the risk of mortality is lower than those with diabetes mellitus alone [7]. Many organs may be harmed or fail in individuals with COVID-19 and two or more comorbidities, increasing the likelihood of severe illness and death [12].

So far, there are few studies on the joint effect of diabetes mellitus and hypertension on mortality in COVID-19 patients; even in Indonesia, there has been no similar study. A study conducted in an area with a high prevalence of chronic diseases and high COVID-19 mortality numbers, such as Yogyakarta, was needed. This study aims to investigate the joint effect between diabetes mellitus and hypertension on in-hospital mortality among COVID-19 patients in Yogyakarta, stratified by age groups and other comorbidity statuses.

## 2. Materials and Methods

### 2.1. Study Design and Duration

This was a hospital-based cohort retrospective study that was carried out for a duration of 22 months (March 2020 to December 2021).

### 2.2. Study Setting

This study was conducted at two large hospitals in the Sleman district, Sleman Public Hospital and UGM Academic Hospital (RSA UGM) Yogyakarta. Both hospitals are COVID-19 referral hospitals, so both hospitals treated COVID-19 patients from the Sleman district.

### 2.3. Study Population

The population of this study was all COVID-19 hospitalized patients in the Sleman district from March 2020 to December 2021.

### 2.4. Sample Size and Sampling Technique

This study used total sampling with the inclusion criteria: COVID-19 patients treated in Sleman District Public Hospital and UGM Academic Hospital from 1 March 2020 to 31 December 2021, and COVID-19 patients who lived in Sleman. The total number of study participants obtained were 2779 respondents, consisting of 959 COVID-19 patients from Sleman Public Hospital and 1820 patients from UGM Academic Hospital.

### 2.5. Study Tools and Techniques

Data was collected by searching the medical records of COVID-19 patients treated at Sleman Public Hospital and RSA UGM. The data collected included: patient outcome (discharged alive or deceased), gender (male or female), age, COVID-19 wave, patient comorbidities, and length of stay. Age was categorized into 10-year intervals in this study. The COVID-19 wave was divided into the Delta wave (1 April 2021 to 31 December 2021) and non-Delta wave (1 March 2020 to 31 March 2021) based on data from the Sleman district health office. We collected data about patient comorbidities, such as diabetes (DM), hypertension (HT), heart disease, kidney disease, asthma, stroke, obesity, chronic pulmonary obstructive disease (COPD), HIV/AIDS, and tuberculosis. The comorbidities data was collected based on diagnosis on medical records. Length of stay was calculated from the first day of hospitalization until the time when patients were out of hospital (recovered) or death.

### 2.6. Statistical Analysis

First, we compared the distribution of patient characteristics and comorbidities across study outcomes (discharged alive or deceased). As in our previous study [13,14], we utilized a *p*-value less than 0.2 from simple binary logistic regression model as the cut-off point to include independent variables into multivariate analysis using the multiple logistic regression model. The multiple binary logistic regression model was used to calculate adjusted odds ratios (AORs) between potential risk factors of COVID-19 and in-hospital mortality. We used Cramer’s V to assess multicollinearity between independent variables. This study used 5% as alpha which corresponded with a 95% confidence level. Our main analysis was the joint effect between DM and HT. The joint effect was estimated by using approach from Strengthening the Reporting of Observational Studies in Epidemiology (STROBE) guidelines [15]. For our joint effect, we coded a four-level variable, indicating patients without both comorbidities, patients with HT but no DM, patients with DM but no HT, and patients with both comorbidities.

## 3. Results

Table 1 presents the characteristic of COVID-19 patients included in this study. There were a total of 369 (13.28%) COVID-19 patients that died in the hospital. As many as 657 (23.64%) and 752 (27.06%) COVID-19 patients had diabetes mellitus (DM) and hypertension (HT), respectively. In this study, we also found that 118 (35.65%) hospitalized COVID-19 patients had both DM and HT. Male COVID-19 patients (49.01%) were of a similar number to female COVID-19 patients (50.99%). Most of the COVID-19 patients treated in the hospital were aged 51 years or above (53.29%). Among all age groups, cases of age 71 or more had the highest mortality rate (26.12%), followed by age 61–70 years (19.34%) and 51–60 years (15.06%). The mortality rate of cases with DM and HT were 27.70% and 25.93%, respectively. Compared with those discharged alive, patients who died had a shorter length of stay. Among other comorbidities, stroke (40.71%) and kidney disease (36.36%) had the highest in-hospital mortality rate.

Table 2 lists the results of simple and multiple logistic regression models for COVID-19 mortality rates in Yogyakarta. There were two comorbidities not included in the final multiple logistic models, which are asthma and HIV. Patients aged 71+ years had the highest risk of in-hospital mortality (adjusted odds ratio [AOR]: 20.09; 95% CI: 4.76–84.75) among other age groups. Compared with COVID-19 patients without DM, COVID-19 patients with DM had approximately three times (AOR: 2.93; 95% CI: 2.22–3.86) higher risk of in-hospital mortality. Patients with hypertension were two times (AOR: 2.06; 95% CI: 1.56–2.73) more likely to have in-hospital mortality than those without hypertension. Other risk factors for in-hospital mortality were age 61–70 years (AOR: 14.19; 95% CI: 3.37–59.82), age 51–60 years (AOR: 12.75; 95% CI: 3.03–53.64), age 41–50 years (AOR: 11.66; 95% CI: 2.73–49.85), age 31–40 years (AOR: 6.82; 95% CI: 1.55–30.01), age 21–30 years (AOR: 6.67; 95% CI: 1.46–30.49), having kidney disease (AOR: 3.22; 95% CI: 2.11–4.90), having stroke (AOR: 2.98; 95% CI: 1.86–4.76), having obesity (AOR: 2.36; 95% CI: 1.10–5.07), having COPD (AOR: 3.11; 95% CI: 1.51–6.39), having tuberculosis (AOR: 2.63; 95% CI: 1.25–5.57), and patients with a length of stay less than or equal to four days (AOR: 10.98; 95% CI: 8.86–13.61).

Table 3 provides the result of the joint effect between DM and hypertension on all age groups, stratified by age 0–40 years and 41–71+ years. The result showed that there is a significant joint effect of DM and hypertension in all groups. COVID-19 patients who had both DM and hypertension were three times (AOR: 3.21; 95% CI: 2.45–4.19) more likely to have in-hospital mortality than those without both comorbidities. Among those aged 0–40 years, the risk for in-hospital mortality was 16 times (AOR: 16.03; 95% CI: 6.01–42.84) higher in COVID-19 patients with both DM and hypertension compared with those without both comorbidities, while among COVID-19 patients aged 41–71+ years, patients with both DM and hypertension had higher risk (AOR: 2.94; 95% CI: 2.23–3.87) of experiencing in-hospital mortality than those without both comorbidities. The highest risk of in-hospital mortality was found in COVID-19 patients without other comorbidities (other than DM and HT) and younger age (age 0–40 years) with AOR equal to 22.40 (95% CI: 6.61–75.99).

## 4. Discussion

In this study, we found that the proportion of hospitalized COVID-19 patients with DM, HT, and both comorbidities were 23.64%, 27.06%, and 35.65%, respectively. This number is higher than those reported in previous studies which ranged from 2.02% to 17.65% [7,16,17,18,19,20,21]. A meta-analysis conducted by Emami et al. showed that the pooled prevalence of diabetes mellitus among COVID-19 patients was 7.87% [22]. Another meta-analysis study reported that the weighted prevalence of diabetes among COVID-19 patients was 20% [23]. This number also higher than the number of DM and HT prevalence in Yogyakarta province and Indonesia which were 11.01% and 8.8%, respectively [24]. Finally, our study also finds that the prevalence of both DM and HT among COVID-19 patients were 11.91%, which is higher than other studies [7].

One of the most notable finding in this study is an interaction between diabetes mellitus and hypertension with COVID-19 and in-hospital mortality. Our finding corroborates with previous findings by Sun et al. [7] that the combined effect of diabetes mellitus and hypertension increase the risk of mortality among COVID-19 patients. This finding may be due to COVID-19 patients with DM already having higher risks of severe complication, and this condition being further aggravated by HT. Public health interventions should focus on adults with both DM and HT. People with both DM and HT should become the highest priority to get vaccinated, including booster doses. The government also needs to educate people to undergo a community health screening about DM and HT. Lastly, a targeted health education should be given to people with both comorbidities.

This study reported similar results with previous studies, that COVID-19 patients with diabetes mellitus had higher risk of in-hospital mortality than those without diabetes mellitus [5,6,7,8,23,25,26,27,28,29]. This result could be explained by chronic inflammatory conditions occurring among COVID-19 patients with DM. COVID-19 patients with DM have significantly higher inflammatory markers, such as interleukin 6 (IL-6), C-reactive protein (CRP), and neutrophil–lymphocyte ratio (NLR) than those without DM [16,30,31,32,33]. Those inflammatory markers were associated with higher risk of death among COVID-19 patients [34,35,36,37]. Interventions related to controlling blood glucose among DM patients and screening for earlier DM diagnosis were needed to reduce the risk of death caused by DM among COVID-19 patients.

Our finding corroborates with previous studies that COVID-19 patients with hypertension had a higher risk of in-hospital mortality than those without hypertension [5,6,7,8,9,10,25,27]. The exact mechanism of this finding remains unclear, but some studies have hypothesized that it may be due to an increase in inflammatory markers. Results from the BRACE-CORONA trial showed that IL-10 and IL-12 levels could predict the severity of COVID-19 among hypertensive patients [38].

This research has several limitations. First, our research is limited to two hospitals in the Sleman area of Yogyakarta, Indonesia. To increase the generalizability of future research, additional hospitals with varying levels of care should be included in future studies. Second, our study accounts only for regions with a high prevalence of non-communicable diseases. The outcome may vary in regions with a lower frequency of non-communicable diseases. Third, due to the limited data available in medical records, this study was unable to collect data on crucial variables such as immunization status, variants, controlled DM/HT, and other crucial variables. At the time of study, Indonesia already had a vaccination campaign, but vaccination status is not well-documented in hospital records.

## 5. Conclusions

This study identified a joint effect between diabetes mellitus and hypertension which increases the risk of in-hospital mortality among COVID-19 patients. We also identified several significant risk factors, such as: age, gender, kidney disease, tuberculosis, COPD, obesity, stroke, heart disease, COVID-19 wave, and length of stay. Targeted public health, clinical, and health education interventions should be carried out on individuals with diabetes mellitus and/or hypertension, such as early diagnosis of diabetes mellitus, health education among elderly people, and vaccination priority for those with diabetes mellitus and/or hypertension.

## Figures and Tables

**Table 1 healthcare-10-02103-t001:** Characteristics of study participants.

Characteristics	Deceased *n* (%)	Discharged Alive *n* (%)	Total *n* (%)
Case’s age	59.93 ± 15.25	46.52 ± 20.98	48.31 ± 20.81
Case’s age group			
71+ years 61–70 years 51–60 years 41–50 years 31–40 years 21–30 years 11–20 years 0–10 years	93 (26.12) 99 (19.34) 92 (15.06) 45 (11.19) 23 (6.50) 14 (4.91) 1 (1.89) 2 (0.98)	263 (73.88) 413 (80.66) 519 (84.94) 357 (88.81) 331 (93.50) 271 (95.09) 52 (98.11) 202 (99.02)	356 (12.82) 512 (18.44) 611 (22.00) 402 (14.48) 354 (12.75) 285 (10.26) 53 (1.91) 204 (7.35)
Case’s gender			
Male Female	196 (14.39) 173 (12.21)	1166 (85.61) 1244 (87.79)	1362 (49.01) 1417 (50.99)
Diabetes mellitus (DM)			
Yes No	182 (27.70) 187 (8.81)	475 (72.30) 1935 (91.19)	657 (23.64) 2122 (76.36)
Hypertension (HT)			
Yes No	195 (25.93) 174 (8.58)	557 (74.07) 1853 (91.42)	752 (27.06) 2027 (72.94)
Heart disease			
Yes No	46 (30.07) 323 (12.30)	107 (69.93) 2303 (87.70)	153 (5.51) 2626 (94.49)
Kidney disease			
Yes No	56 (36.36) 313 (11.92)	98 (63.64) 2312 (88.08)	154 (5.54) 2625 (94.96)
Asthma			
Yes No	12 (13.48) 357 (13.27)	77 (86.52) 2333 (86.73)	89 (3.20) 2690 (96.80)
Stroke			
Yes No	46 (40.71) 323 (12.12)	67 (59.29) 2343 (87.88)	113 (4.07) 2666 (95.93)
Obesity			
Yes No	12 (26.67) 357 (13.06)	33 (73.33) 2377 (86.94)	45 (1.62) 2734 (98.38)
COPD			
Yes No	15 (31.91) 354 (12.96)	32 (68.09) 2378 (87.04)	47 (1.69) 2732 (98.31)
HIV			
Yes No	2 (22.22) 367 (13.25)	7 (77.78) 2403 (86.75)	9 (0.32) 2770 (99.68)
Tuberculosis			
Yes No	11 (21.15) 358 (13.13)	41 (78.85) 2369 (86.87)	52 (1.87) 2727 (98.13)
COVID-19 Wave			
Delta Wave Non-Delta Wave	306 (14.32) 63 (9.81)	1831 (85.68) 579 (90.19)	2137 (76.90) 642 (23.10)
Length of stay (days)	6.98 ± 5.97	8.23 ± 4.09	8.06 ± 4.40
Length of stay category			
≤4 days >4 days	174 (35.51) 195 (8.52)	316 (64.49) 2094 (91.48)	490 (17.63) 2289 (82.37)
Hospitals			
Sleman District Public Hospital UGM Academic Hospital	131 (13.66) 238 (13.08)	828 (86.34) 1582 (86.92)	959 (34.51) 1820 (65.49)

**Table 2 healthcare-10-02103-t002:** Bivariate and multivariate analysis using simple and multiple logistic regression models.

Characteristics	Bivariate Analysis		Multivariate Analysis	
Case’s age group	OR	95% CI	AOR	95% CI
71+ years 61–70 years 51–60 years 41–50 years 31–40 years 21–30 years 11–20 years 0–10 years	35.72 24.21 17.90 12.73 7.02 5.22 1.94 1	8.70–146.68 5.91–99.16 4.37–73.35 3.06–53.03 1.64–30.08 1.17–23.22 0.17–21.84	20.09 14.19 12.75 11.66 6.82 6.67 2.87 1	4.76–84.75 3.37–59.82 3.03–53.64 2.73–49.85 1.55–30.01 1.46–30.49 0.25–33.59
Case’s gender				
Male Female	1.21 * 1	0.96–1.51	0.96 1	0.74–1.25
Diabetes mellitus				
Yes No	3.96 1	3.13–5.00	2.93	2.22–3.86
Hypertension				
Yes No	3.73 1	2.95–4.69	2.06 1	1.56–2.73
Heart disease				
Yes No	3.06 1	2.07–4.46	1.74 1	1.13–2.67
Kidney disease				
Yes No	4.22 1	2.92–6.05	3.22 1	2.11–4.90
Asthma				
Yes No	1.02 1	0.49–1.90		
Stroke				
Yes No	4.98 1	3.28–7.49	2.98 1	1.86–4.76
Obesity				
Yes No	2.42 1	1.13–4.86	2.36 1	1.10–5.07
COPD				
Yes No	3.15 1	1.57–6.05	3.11 1	1.51–6.39
HIV				
Yes No	1.87 1	0.19–9.85		
Tuberculosis				
Yes No	1.78 1	0.82–3.56	2.63 1	1.25–5.57
COVID-19 Wave				
Delta Wave Non-Delta Wave	1.54 1	1.15–2.08	1.27 1	0.92–1.77
Length of stay category				
≤4 days >4 days	5.91 1	4.63–7.53	8.85 1	6.67–11.73
Hospitals				
Sleman District Public Hospital UGM Academic Hospital	1.05 1	0.84–1.32		

* *p*-value < 0.20.

**Table 3 healthcare-10-02103-t003:** Interaction result between diabetes mellitus and hypertension on COVID-19 in-hospital mortality stratified with age group.

Interaction Term ^a^	AOR	*p*-Value	95% CI
All age group Have DM and HT Have DM but not HT Have HT but not DM Age 0–60 years Have DM and HT Have DM but not HT Have HT but not DM Age 61–71+ years Have DM and HT Have DM but not HT Have HT but not DM No other comorbidities and aged 0–60 years Have DM and HT Have DM but not HT Have HT but not DM No other comorbidities and aged 61–71+ years Have DM and HT Have DM but not HT Have HT but not DM	5.99 3.25 2.27 9.77 4.68 1.93 4.10 2.06 2.45 12.71 6.72 2.54 6.35 3.41 3.52	<0.001 <0.001 <0.001 <0.001 <0.001 0.026 <0.001 0.014 <0.001 <0.001 <0.001 0.003 <0.001 0.001 <0.001	4.19–8.59 2.21–4.79 1.57–3.28 5.88–16.23 2.76–7.94 1.08–3.45 2.45–6.86 1.15–3.67 1.48–4.06 6.84–23.59 3.57–12.64 1.21–5.33 3.21–12.56 1.71–6.82 1.82–6.81

^a^ adjusted for gender, kidney disease, stroke, COPD, and length of stay at hospitals.

## Data Availability

The dataset for this study is not publicly available.

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
