# Peer review of "Joint Effect of Diabetes Mellitus and Hypertension on COVID-19 in-Hospital Mortality Stratified by Age Group and Other Comorbidities: A Cohort Retrospective Study Using Hospital-Based Data in Sleman, Yogyakarta"

_healthcare, 2022, doi:10.3390/healthcare10102103_

Round 1
Reviewer 1 Report
The original research study entitled "Joint Effect of Diabetes Mellitus and Hypertension on COVID-19 in-Hospital Mortality Stratified by Age Group and Other Comorbidities: A Cohort Retrospective Study Using Hospital Based Data in Sleman, Yogyakarta" is a well written and interesting manuscript which requires some minor reviews:
> Introduction page 1 line 36: COVID-19 is not a virus, it is a disease. the virus is SARS-CoV-2
> Methods page 2 line 71: Maret should be replaced by March
> Results page 3 line 122: the phrase should be rewritten. The higher mortality in those stayed less than 4 days is precisely due to early mortality so lenth of stay was not a co-variable
> Table 1: you should add the statistical data
> Results page 5 line 134: the phrase should be rewritten 106% more likely is not the right expression
Congratulations for a good paper
Author Response
please the following attached documents for the reviewer response.

Reviewer 2 Report
Thank you for the opportunity to review thiw paper.
The results of this study 2779 COVID-19 patients show that the patients with both DM and HT had a greater in-hospital mortality, especially at younger age (0-40yrs). These results are concordant with other studies results. As you mentioned, these results are useful for the recommandations for the vaccination of this target group (DM+HT/DM/HT), including the booster doses and also for the specific antiviral therapy.
One of the limitations of this study was the poor data related to the immunization state of these patients (patients enrolled from January 2021 to December 2021, when the specific vaccin was disponible). Please include in your discussion some comments about the specific-vaccinated patients.
Also, please specify in the discussion if there is any corellation of the in-hospital mortality with the poor controled DM and/or HT, or the mortality is similar for the patients with well controlled DM/HT.
Author Response
Please see attached documents regarding response to reviewers.
